# Clinical Evaluation of Micro-Embolic Activity with Unexpected Predisposing Factors and Performance of Horizon AF PLUS during Cardiopulmonary Bypass

**DOI:** 10.3390/membranes12050465

**Published:** 2022-04-26

**Authors:** Ignazio Condello, Roberto Lorusso, Giuseppe Santarpino, Flavio Fiore, Giuseppe Nasso, Giuseppe Speziale

**Affiliations:** 1Department of Cardiac Surgery, Anthea Hospital, GVM Care & Research, 70124 Bari, Italy; santarpino@unicz.it (G.S.); ffiore@gvmnet.it (F.F.); gnasso@libero.it (G.N.); gspeziale3@virgilio.it (G.S.); 2Cardio-Thoracic Surgery Department, Heart and Vascular Centre, Maastricht University Medical Centre, 6229 HX Maastricht, The Netherlands; roberto.lorussobs@gmail.com; 3Cardiovascular Research Institute Maastricht, 6229 ER Maastricht, The Netherlands; 4Department of Cardiac Surgery, Paracelsus Medical University, 90419 Nuremberg, Germany; 5Cardiac Surgery Unit, Department of Experimental and Clinical Medicine, “Magna Graecia” University of Catanzaro, 88100 Catanzaro, Italy

**Keywords:** gaseous micro-emboli, cardiopulmonary bypass, horizon venous reservoir, A.L.ONE AF PLUS Oxygenator, GAMPT BCC 300, thermal exchange, oxygenation performance

## Abstract

**Background:** During Cardiopulmonary Bypass (CPB) gaseous micro-emboli (GMEs) decrease the quality of the blood flow and the capillary oxygen delivery, increasing the incidence of postoperative neurocognitive disorders (POCD) following cardiac surgery. In these circumstances, the use of an efficient device, could be crucial for the removal and reduction of micro-embolic activity. **Methods:** From February 2022 to March 2022, we prospectively collected data from 40 consecutive patients undergoing conventional and minimally invasive cardiac surgery that used the Horizon AF PLUS (Eurosets, Medolla, Italy). We collected, during the CPB’s time, the incidence of unexpected predisposing factors for micro-embolic activity reported in the literature with the GMEs count and their diameter through the GAMPT BCC 300 (Germany). **Results:** The group of patients without unexpected predisposing factors for micro-embolic activity (55%) reported a GME volume of 0.59 ± 0.1 (μL) in the arterial line (*p*-value 0.67). In both groups were no reported performance deficit during the procedures for oxygenation and CO_2_ removal. **Conclusions:** Our clinical analysis showed that Horizon AF PLUS is an effective and safe device without iatrogenic perioperative complications, for the reduction of micro embolic activity during CPBs procedures, with high efficiency in terms of oxygenating performance and thermal exchange.

## 1. Introduction

The detection and prevention of gaseous micro-emboli (GMEs) during cardiopulmonary bypass has generated considerable interest within the cardiac surgical community. During cardiopulmonary bypass (CPB) conduction, management methods and many unexpected predisposing factors could generate micro-embolic activity (MEA). GMEs decrease the quality of the blood flow and the capillary oxygen delivery, increasing the incidence of transient postoperative neurocognitive disorders (POCD) following cardiac surgery (i.e., postoperative delirium and agitation after anesthesia discontinuation) [1]. Postoperative delirium is defined as “a clinical situation in which patients think and speak incoherently, are disoriented and show impairment of memory and attention”, which is not explained by a medical history of dementia, but affects the ability to focus, the mechanical ventilation (MV) and the duration and intensive care unit (ICU) length of stay [2]. In these circumstances, the use of an efficient device consisting of two components, the venous reservoir and oxygenator, could be crucial in the removal, and thus reduce MEA. However, no consensus exists on when a given diameter or number of emboli becomes injurious to the patient. An important variable is the gas mixture inside the bubble. Nitrogen has a very long dissolution time that results in prolonged ischemia for tissue behind the occlusion. The pathophysiologic reaction of the body, when exposed to GMEs, is most likely based on ischemia caused by partial occlusion of the blood vessels and by endothelial damage. GMEs can be cleared mechanically by using filters, by a reduction in blood velocity, and by a rapid reduction in the nitrogen content [3]. Elimination of GMEs is dependent on the design of the cardiopulmonary bypass circuit. A membrane oxygenator, although not designed for it, can remove GMEs. Arterial line filtration is not the best solution for the removal of GMEs, because larger emboli have been fractionated before reaching the arterial filter. Venous line filtration is a more efficient way of clearing gaseous micro-emboli [4]. In this context, we conducted an evaluation of micro-embolic activity with unexpected predisposing factors on Horizon AF PLUS during CPB and oxygenation performance.

## 2. Materials and Methods

### 2.1. Population and Study Design

From February 2022 to March 2022, we prospectively collected the data of forty patients for elective cardiac surgery procedures at our institution (Department of Cardiothoracic Surgery, Anthea Hospital, Bari, Italy): isolated aortic valve replacement (*n* = 15), coronary artery bypass graft (*n* = 15) and mitral valve repair with a minimally invasive approach (*n* = 10) (Figure 1). The patients were aged >28 to 80 years, without chronic kidney failure and with a calculated European System for Cardiac Operative Risk Evaluation II score (mean value, 2.1–2.4%). The study protocol was approved by the local ethics committee and all patients provided written consent to the scientific treatment of their data. All 40 patients, for this study, used the Horizon AF PLUS during cardiopulmonary bypass, which consists of a hard-shell cardiotomy/venous reservoir integrated with two cardiotomy filters, designed to allow venous drainage of the patient’s blood, both through the hydrostatic load (height difference between the patient and the reservoir) and the vacuum-assisted venous drainage (VAVD) technique and A.L.ONE AF PLUS (Figure 2). The membrane that the oxygenator uses is a microporous hollow-fiber membrane consisting of a gas exchange module with an integrated heat exchanger and an integrated 38 µm arterial filter that ensures arterial blood filtration with the removal of microaggregates and micro-emboli (Eurosets, Medolla, Italy) (Figure 3) with only a roller pump. HORIZON AF PLUS’s inner contact surfaces are coated with the A.G.I.L.E. (Advanced Generation Inert Layer ECC) system, based on phosphorylcholine (PC), improving the device’s blood compatibility by reducing platelet adhesion on the coated surface. Perioperative data included CPB duration, and the incidence of unexpected predisposing factors for micro-embolic activity reported in the literature: a low level in the venous reservoir (<250 mL), vacuum-assisted venous drainage (VAVD), accidental air embolism from the venous line, excessive suction use from aspirators with the GMEs count and their diameter through the GAMPT BCC 300 (Figure 4 and Figure 5). The probes were positioned: in the venous drainage line, in the outlet line of the venous reservoir and in the arterial line [1,2,3,4,5]. The primary endpoint was the evaluation of the efficiency of the venous reservoir (VR) and oxygenator (Horizon VR and Oxygenator (Oxy) A.L.ONE AF PLUS) in GME removal, in terms of the number of total microbubbles with a diameter and air microliters in the arterial line at the end of CPB in relation to unexpected predisposing factors for micro-embolic activity. The secondary endpoint was the evaluation of oxygenation performance in terms of the mean values of PaO_2_ and PaCO_2_ for the target gas blender of 2.0 ± 0.3 L/min and 55 ± 5% FiO_2_, and the evaluation of efficiency in the maintenance of thermal exchange in mild hypothermia (from 36 °C to 34 °C and vice versa for nasopharyngeal temperature) for the integrated heat exchanger unit in the A.L.ONE AF PLUS Oxygenator. 

### 2.2. Anesthetics and Surgical Procedures

The operation was performed under general anesthesia (using propofol, fentanyl, midazolam and rocuronium) using the SedLine^®^ brain function monitoring system (Masimo Corporation, Irvine, CA, USA). The patient was intubated and anesthetized. The arterial and venous lines were prepared. A single-lumen endotracheal tube was used for pulmonary ventilation. A transesophageal echocardiographic (TEE) probe was inserted to examine the anatomy and morphology of the aortic valve and the ascending aorta, and to evaluate aortic valve function and the removal of air before the removal of the cross-clamp. The adhesive pads of the defibrillator were correctly placed on the thoracic wall. The trigger for the administration of red blood cells (RBC) units was a hemoglobin level of less than 8 g/dL both during CPB and in the ICU. For antagonization of heparin, 0.5–0.75 mg protamine was applied for every 100 heparin units. Aortic valve replacement and coronary artery bypass graft procedures were performed in the median sternotomy with central cannulation, MVR was with the right mini-thoracotomy approach with peripheral cannulation, and surgical procedures were performed as routine by 2 surgeons. Concentrated red blood cells were transfused whenever Hb concentrations fell below 6 g/dL during surgery or below 8 g/dL during an ICU stay [2].

### 2.3. CPB Setting

Only the open system (Horizon VR and Oxy AF PLUS, Eurosets, Medolla, Italy) was used for CPB. All patients were treated with mild hypothermic CPB (34–36 °C); a volume of 1250 mL crystalloid Ringer acetate solution was used for priming. The surgical procedures selected for this study do not justify the use of moderate hypothermia by falling below 34 °C. For this reason, in the event of an initial increase in anaerobic metabolism, the first compensation approach was not to lower the temperature; however, possibly liquids or red blood cells were integrated. The hardware consisted of a Stöckert S5 heart-lung machine and a Stöckert 3T heater-cooler system (LivaNova), and the same cannulae were employed in both groups. The venous drainage line (3/8 inch) and the arterial delivery line (3/8) were each 180cm long, the lines to the pump (3/8 and 1/2) were each 80 cm, and the cardioplegia line (1/16) was 190 cm. The aspiration lines were 1/4. This circuit uses a serial pump with VAVD. Roller pumps were used because aspiration has a management nadir below from 800 mL/min to >2 L/min. A negative pressure of −40 mmHg VAVD was applied to the reservoir. The intracavitary aspirator managed with a roller pump was channeled into a venous reservoir, and the extra-cavitary aspirator was managed with a roller pump [1].

The landing monitoring system (Eurosets) was used for DO_2_ management during CPB. Metabolic parameters were monitored with a DO2 system; the nadir was higher than 280 mL/min/m^2^. The security system used a level alarm, and a bubble probe was used to detect microbubbles leaving the venous reservoir. Anticoagulant therapy consisted of heparin sodium before CPB at 300 IU/kg to give an ACT of greater than 4 by 80 s. Cardioplegia was performed in an antegrade manner with normothermic blood in a 190 cm closed circuit with a bubble-trap filter by a serial micrometric pump, with St. Thomas solution with procaine and repeated every 30 min [2]. During the CPBs procedures, the oxygenator filter purge was kept and managed closed. The GAMPT system, the BCC300, was used for GMEs’ count during the procedures. The BC300 uses a pulsed ultrasonic Doppler system with a transmission frequency of 2 MHz. From the Doppler signal of a bubble, one obtains an amplitude-modulated low-frequency signal depending on the size of the bubble and the time in the sound field of the sensor. By means of different filter functions and Hilbert transformations, the signal envelope was calculated and corrected by the reference signal. The maximum amplitude of the corrected signal was a measure of the bubble size. According to the manual, the BC300 is capable of measuring GME between 5 and 500 mm. The detection limit is 1000 GME per second and it can be used with blood flows between 0.5 and 8 L/min.

## 3. Statistical Analysis

Continuous data were expressed as mean ± standard deviation or a median with the interquartile range and categorical data as percentages. Cumulative survival was evaluated with the Kaplan–Meier method. All reported *p*-values were two-sided, and *p*-values of <0.05 were considered to indicate statistical significance. All statistical analyses were performed with SPSS 22.0 (SPSS Inc., Chicago, IL, USA).

## 4. Results

The mean age was 67 ± 15 years, 40 patients underwent cardiac surgery for conventional and minimally invasive cardiac surgery. Demographic, preoperative and operative details of the patient population are shown in Table 1 and Table 2. The unexpected predisposing factors for micro-embolic activity were reported in 18 patients (45%) who underwent CPB procedures, (*n* = 4) accidental air embolism from venous line time (mean values 3 ± 1 min); (*n* = 10) low levels in the venous reservoir (<250 mL) in association with vacuum-assisted venous drainage (≥40 mmHg) time (mean values 8 ± 5 min); (*n* = 4) report a combination of two excessive suctions of aspirators (>2 L/min, >77 RPM) with low levels in the venous reservoir (<250 mL) in association with vacuum-assisted venous drainage (≥40 mmHg) time (mean values 10 ± 2 min). Twenty-two patients (55%) did not report unexpected predisposing factors for micro-embolic activity (Table 3). The patients that reported unexpected predisposing factors for micro-embolic activity reported mean values: n° of bubbles was 259.766 ± 193, 33% reported a diameter >500 μm, GME volume was 40.9 ± 2 (μL) in the venous inlet line of the venous reservoir; n° of bubbles was 68.053 ± 51, 2.3% reported a diameter >500 μm; GME volume was 1.31 ± 0.3 (μL) in the outlet line of the venous reservoir after pump; n° of bubbles was 1.045 ± 41, 0% reported a diameter >500 μm; GME volume was 0.71 ± 0.2 (μL) in the arterial line. The group of patients without unexpected predisposing factors for micro-embolic activity (55%) reported a GME volume of 0.59 ± 0.1 (μL) in the arterial line (*p*-value 0.67) (Table 4 and Table 5). Mean values of PaO_2_ in both groups were 260 mmHg ± 25, and PaCO_2_ was 38 mmHg ± 4, with no reported performance deficit during the procedures for oxygenation and CO_2_ removal (Table 6). The desired thermal objectives in the management of mild hypothermia during CPBs procedures were reached from 36 to 34 °C for the nasopharyngeal temperature (mean time of 5.5 min) by setting a temperature of 34 °C in the heater-cooler device (HCD) and vice versa (mean time 6.5 min), by setting a temperature in the HCD of 36.5 °C (Figure 6) (Table 7).

## 5. Discussion

GMEs are considered a cause of neurocognitive deficits. The pathophysiological mechanism is multiple [3]. When a microbubble occludes a blood vessel, hypoxia will occur downstream from the blockage [5]. The duration of hypoxia and the deleterious effects of this hypoxia will very much depend on the size and number of GME as well as on the gas composition of these microbubbles [6]. At the same time, the microbubble will induce local inflammation with edema formation, which will increase the cerebral area at risk. Finally, reperfusion injury may cause additional harm once the initial ischemia is resolved [7]. Microbubbles between 20 and 60 μm will be distributed according to the blood flow since they have virtually no buoyancy in moving blood. Microbubbles of 20 μm will be absorbed quickly, depending upon the solubility coefficient and partial pressures of the gas in the bubble and the surrounding blood and tissue [8]. This occurs if the bubble is not coated with lipid, which changes the rate of gas exchange between the bubble and blood [9]. In this context, for the first time, we have highlighted in our sample an incidence of 45% of sudden events during the conduct of the CPB that can increase the production of GME, independent of the perfusion methodology and technique but linked to the aspect of surgical management [10]. The high efficiency of the Horizon AF PLUS device, in counteracting micro-embolic activity, made the result in terms of the volume of microbubbles in the arterial line at the end of CPB, almost comparable to the group (55%) that did not report predisposing factors for the micro-embolic activity during CPB. The limitations of this study were not related to postoperative outcomes, such as (postoperative neurocognitive disorders (POCD), length of stay in ICU and postoperative quality of life) as both groups in this setting reported a low volume of gas micro-emboli in the arterial line of the CPB. However, aspects related to the gaseous micro-embolic activity are often related in cardiac surgery to the management of de-airing before and after removal of the cross-clamp, and to the CO_2_ management technique in the surgical field. The other limitation of this study was that we analyzed patients with coronary artery disease who are known to have atheroemboli, which could contribute to GME [11,12,13,14,15].

## 6. Conclusions

The patients who used Horizon VR and Oxy A.L.ONE AF PLUS (Horizon AF PLUS, Eurosets, Medolla, Italy) with unexpected predisposing factors for micro-embolic activity were not reported as statistically significant in their difference in terms of (μL) the GME volume in the arterial line at the end of the procedures compared with the group that did not report unexpected predisposing factors for micro-embolic activity. Our clinical analysis showed that Horizon AF PLUS is an effective and safe device without iatrogenic perioperative complications, for the reduction of micro embolic activity during CPBs procedures, with high efficiency in terms of oxygenating performance and thermal exchange. However, further studies and samples are needed to validate this report.

## Figures and Tables

**Figure 1 membranes-12-00465-f001:**
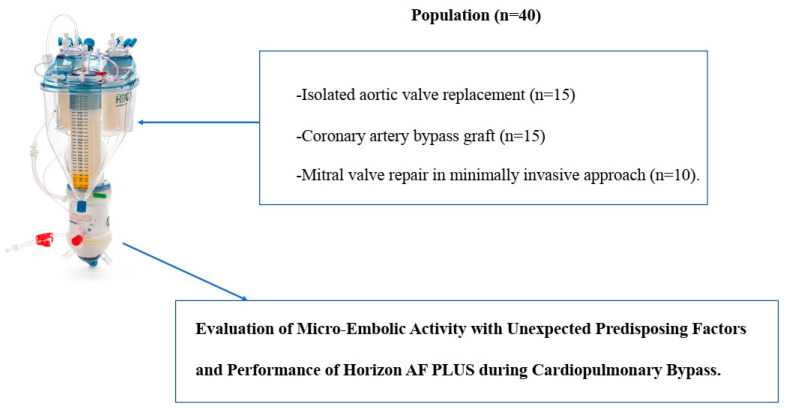
Study population.

**Figure 2 membranes-12-00465-f002:**
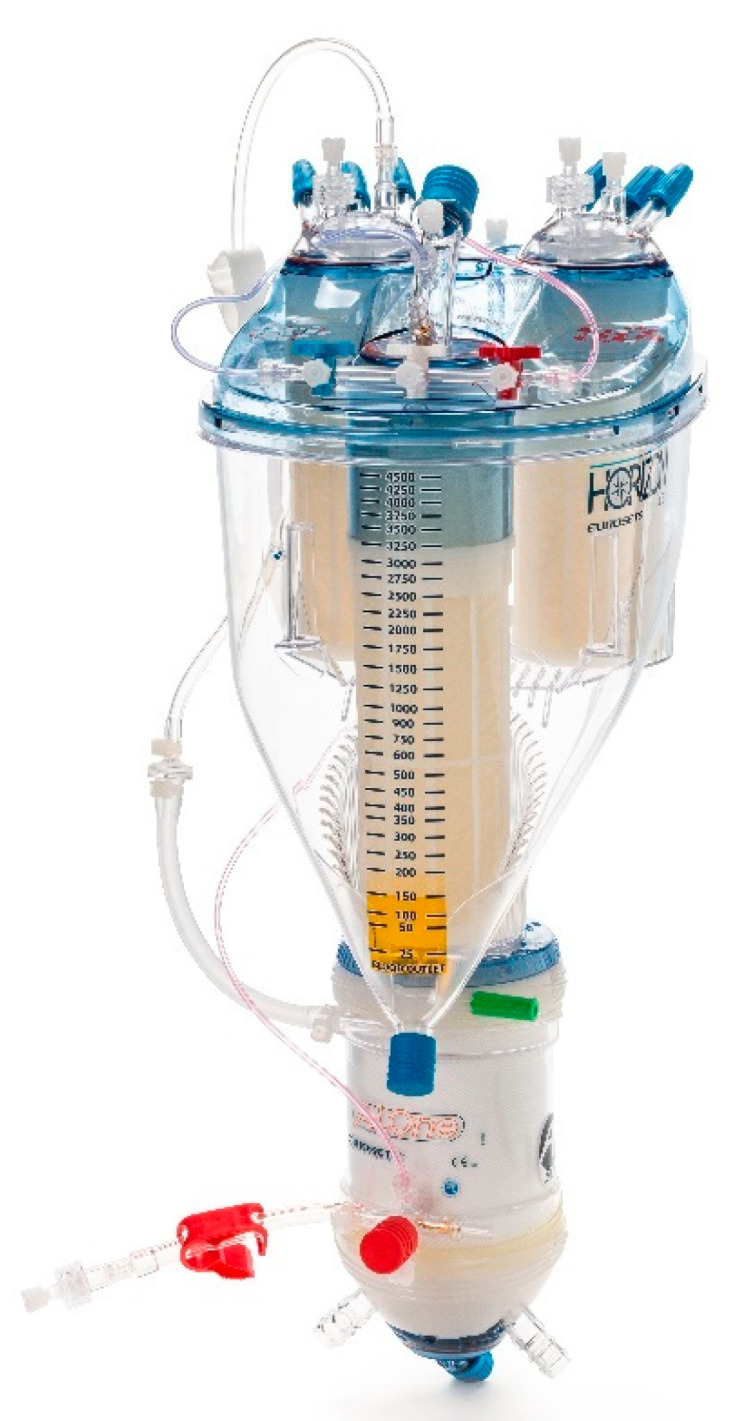
Horizon AF PLUS, Eurosets (venous reservoir and oxygenator).

**Figure 3 membranes-12-00465-f003:**
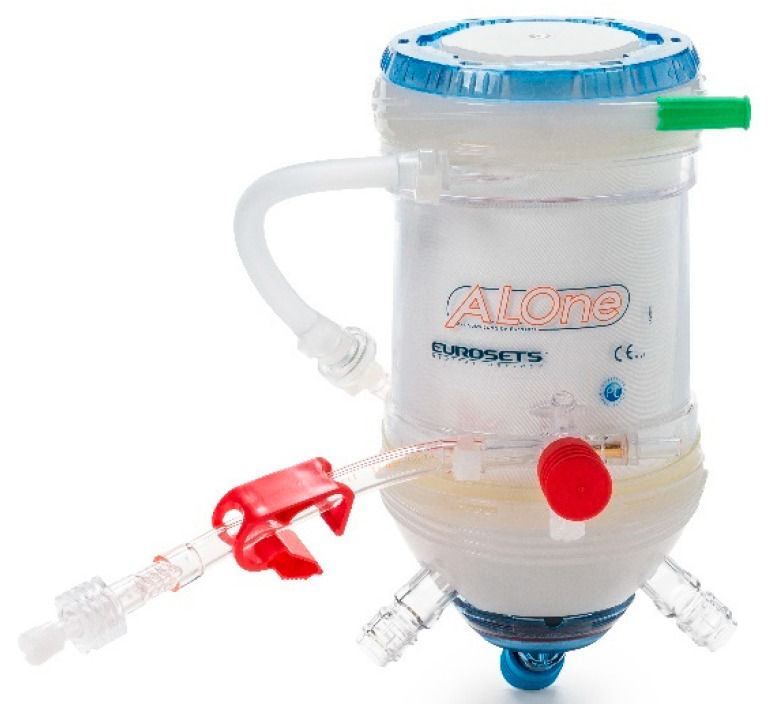
A.L.ONE AF PLUS Oxygenator.

**Figure 4 membranes-12-00465-f004:**
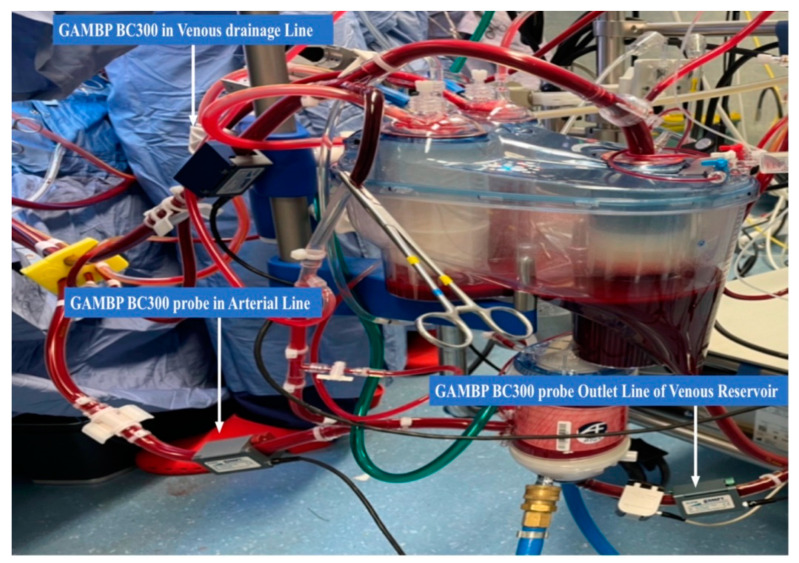
Probes GAMPT BC300 positioning for micro-embolic activity evaluation on Horizon AF PLUS.

**Figure 5 membranes-12-00465-f005:**
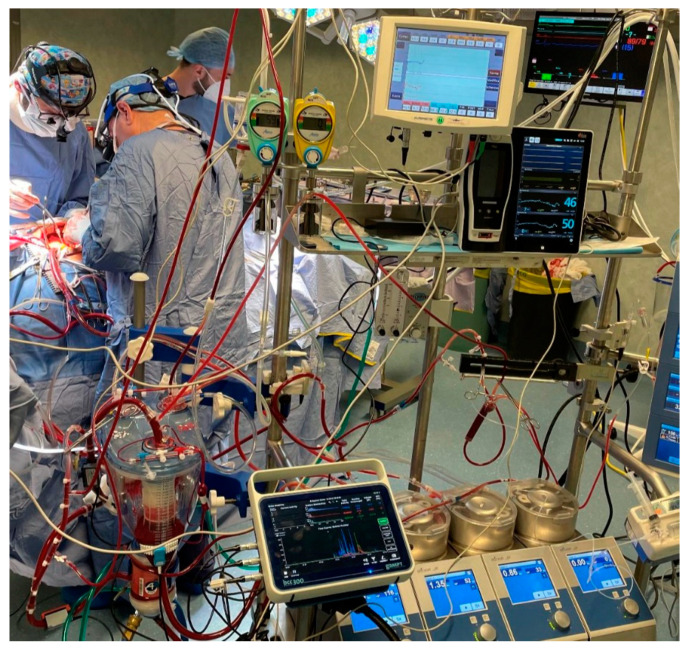
Horizon AF PLUS during cardiopulmonary bypass use with GAMPT BC300.

**Figure 6 membranes-12-00465-f006:**
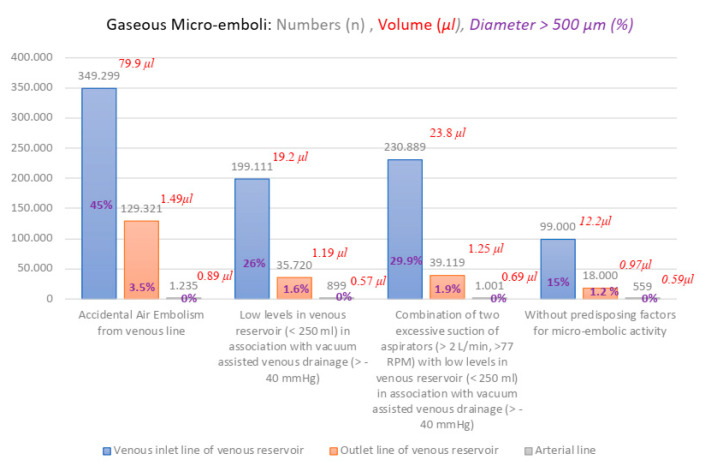
Graphic Representation of gaseous micro-emboli in: Numbers (*n*), Volume (μL), Diameter > 500 μm (%); with and without predisposing factors for micro-embolic activity during Horizon AF PLUS use.

**Table 1 membranes-12-00465-t001:** Preoperative profile and operative data.

Characteristic	Conventional Cardiac Surgery (*n* = 30)	Minimally Invasive Mitral Valve Repair (*n* = 10)
Mean age (y)	69.9	72.5
Male sex	15 (50)	6 (60)
Mean body surface area (m^2^)	1.73	1.78
Mean left ventricular ejection fraction (%)	45	50
Median NYHA functional class	2	2
EuroSCORE II (mean)	2.1	2.4
Pre-CPB hematocrit (%)	34.4 ± 1.2	34.4 ± 1.7
Pre-CPB Hb (g/dL)	10.4 ± 1.1	10.8 ± 1.2
No. of chronic obstructive pulmonary disease cases (mean)	27	28
Creatinine (mg/dL)	1.11 ± 0.4	1.09 ± 0.5
Obstructive coronary artery disease (%)	15	0

Values are presented as *n* (%) or mean ± standard deviation. NYHA, New York Heart Association; EuroSCORE, European System for Cardiac Operative Risk Evaluation; CPB, cardiopulmonary bypass; Hb, hemoglobin.

**Table 2 membranes-12-00465-t002:** Operative data.

Parameter	Conventional Cardiac Surgery (*n* = 30)	Minimally Invasive Mitral Valve Repair (*n* = 10)	*p*-Value
CPB time (min)	104 ± 11.1	102 ± 9.34	0.92
Aortic cross-clamp time (min)	78 ± 5	44 ± 6	0.75
Nadir temperature (°C) during CPB	34.9 ± 1.1	34.7 ± 2.1	0.75
Nadir hemoglobin value (mg/dL) during CPB	8.73 ± 1.53	8.6 ± 1.25	0.88
Nadir hematocrit (%) during CPB	26.6 ± 3.4	26.3 ± 3.9	0.89
Nadir DO_2i_ (mL/min/m^2^) during CPB	294 ± 29	289 ± 14	0.99
O_2_ER_i_ (%) during CPB	23 ± 1	23 ± 5	0.89
Nadir CI (L/min/m^2^) during CPB	2.5 ± 0.2	2.5 ± 0.1	0.91
Nadir SvO_2_ (%)	81 ± 2	80 ± 5	0.93

Values are presented as mean ± standard deviation. CPB, cardiopulmonary bypass.

**Table 3 membranes-12-00465-t003:** Incidence of micro-embolic activity for procedures and duration.

Unexpected Predisposing Factors for Micro-Embolic Activity *n* = 18 (45%)	Conventional Cardiac Surgery (*n* = 30)	Minimally Invasive Mitral Valve Repair (*n* = 10)	Phenomena Duration (min) Mean Values
Accidental air embolism from venous line (*n* = 4)	1	3	3 ± 1
Low levels in venous reservoir (<250 mL) in association with vacuum-assisted venous drainage (≥40 mmHg) (*n* = 10)	8	2	8 ± 5
Combination of two excessive suctions of aspirators (>2 L/min, >77 RPM) with low levels in venous reservoir (<250 mL) in association with vacuum-assisted venous drainage (≥40 mmHg) (*n* = 4)	3	1	10 ± 2
Absence of predisposing factors for micro-embolic activity *n* = 22 (55%)	18	4	0

Values are presented as *n* (%) or mean ± standard deviation. CPB, cardiopulmonary bypass.

**Table 4 membranes-12-00465-t004:** Quantification of micro-embolic activity in the circuit for unexpected predisposing factors for micro-embolic activity on Horizon AF PLUS.

Unexpected Predisposing Factors for Micro-Embolic Activity on Horizon AF PLUS (*n* = 18)	Venous Inlet Line of Venous Reservoir	Outlet Line of Venous Reservoir	Arterial Line
Accidental Air Embolism from venous line (*n* = 4)			
Gaseous micro-emboli numbers	349.299 ± 28	129.321 ± 60	1.235 ± 73
Diameter >500 μm (%)	45	3.5	0
Volume (μL)	79.9 ± 2	1.49 ± 5	0.89 ± 1
Low levels in venous reservoir (<250 mL) in association with vacuum-assisted venous drainage (≥40 mmHg) (*n* = 10)			
Gaseous micro-emboli numbers	199.111 ± 76	35.720 ± 40	899 ± 47
Diameter > 500 μm (%)	26	1.6	0
Volume (μL)	19.2 ± 2	1.19 ± 6	0.57 ± 1
Combination of two excessive suctions of aspirators (>2 L/min, >77 RPM) with low levels in venous reservoir (<250 mL) in association with vacuum-assisted venous drainage (≥40 mmHg) (*n* = 4)			
Gaseous micro-emboli numbers	230.889 ± 100	39.119 ± 25	1.001 ± 37
Diameter >500 μm (%)	29	1.9	0
Volume (μL)	23.8 ± 2	1.25 ± 3	0.69 ± 2
Without predisposing factors for micro-embolic activity on Horizon AF PLUS (*n* = 22)			
Gaseous micro-emboli numbers	99.000 ± 35	18.000 ± 65	559 ± 56
Diameter >500 μm (%)	15	1.2	0
Volume (μL)	12.2 ± 3	0.97 ± 5	0.59

Values are presented as mean ± standard deviation.

**Table 5 membranes-12-00465-t005:** Difference for gaseous micro-emboli volume (μL) in arterial line at the end of CPBs groups with Horizon AF PLUS.

	Unexpected Predisposing Factors for Micro-Embolic Activity on Horizon AF (*n* = 18)	Without Predisposing Factors for Micro-Embolic Activity on Horizon AF (*n* = 22)	*p*-Value
Gaseous micro-emboli volume (μL) in arterial line at the end of CPB	0.71 ± 0.2	0.59 ± 0.1	0.67

Values are presented as mean ± standard deviation. CPB, cardiopulmonary bypass.

**Table 6 membranes-12-00465-t006:** Evaluation of oxygenation performance in terms of mean values of PaO_2_ and PaCO_2_ for the target gas blender of 2.0 ± 0.3 L/min and 55 ± 5% FiO_2_ during CPBs with Horizon AF PLUS.

	PaO_2_ mmHg	PaCO_2_ mmHg
Unexpected predisposing factors for micro-embolic activity on Horizon AF PLUS (*n* = 18)	260 ± 25	38 ± 4
Without predisposing factors for micro-embolic activity on Horizon AF PLUS (*n* = 22)	260 ± 23	38 ± 6

Values are presented as mean ± standard deviation. CPB, cardiopulmonary bypass.

**Table 7 membranes-12-00465-t007:** Thermal exchange performance for mild hypothermia during CPBs with Horizon AF PLUS.

Nasopharyngeal Temperature	From 36 °C to 34 °C HCD Setting 34 °C (min)	From 34 °C to 36 °C HCD Setting 36.5 °C (min)
Unexpected predisposing factors for micro-embolic activity on Horizon AF PLUS (*n* = 18)	5.5 ± 2	6.5 ± 1
Without predisposing factors for micro-embolic activity on Horizon AF PLUS (*n* = 22)	5.5 ± 1	6.5 ± 2

Values are presented as mean ± standard deviation. HCD, heater-cooler device; CPB, cardiopulmonary bypass.

## Data Availability

The datasets analyzed during the current study are available from the corresponding author on reasonable request.

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
