# Peer review of "Clinical Evaluation of Micro-Embolic Activity with Unexpected Predisposing Factors and Performance of Horizon AF PLUS during Cardiopulmonary Bypass"

_membranes, 2022, doi:10.3390/membranes12050465_

Round 1
Reviewer 1 Report
The authors present their results about unexpected Predisposing Factors and the Performance of Horizon AF PLUS during Cardiopulmonary Bypass. They present succinct data but they need to clarify why was the study period chosen to be a shot one Feb 2022 to March 2022.
Where the baseline characteristics matched between the groups, as patients with coronary artery disease are known to have atheroemboli which could contribute to GME.
Finally, they can refer to the paper about GME and air embolism during extracorporeal circulation "Kumar A, Keshavamurthy S, Abraham JG, Toyoda Y. Massive Air Embolism Caused by a Central Venous Catheter During Extracorporeal Membrane Oxygenation. J Extra Corpor Technol. 2019 Mar;51(1):9-11. PMID: 30936582; PMCID: PMC6436163" and cite in their manuscript.
Author Response
Dear reviewer thank you very much for your support and advice,
We used this one-month period from February to March for the study because we had the availability to use the microembolic activity monitoring systems (GAMPT BCC300) for only 40 days and we tried to do the best we could.
In the discussions and limitations we will insert that patients with coronary artery disease are known to have atheroemboli which could contribute to GME.
And we will insert the bibliographic reference about GME and air embolism during extracorporeal circulation "Kumar A, Keshavamurthy S, Abraham JG, Toyoda Y. Massive Air Embolism Caused by a Central Venous Catheter During Extracorporeal Membrane Oxygenation. J Extra Corpor Technol. 2019 Mar; 51 ( 1): 9-11. PMID: 30936582; PMCID: PMC6436163 "and cite in their manuscript.
Reviewer 2 Report
This paper reports on the Clinical Evaluation of Micro-Embolic Activity with Unexpected Predisposing Factors and Performance of Horizon AF PLUS during Cardiopulmonary Bypass. Some results are interesting. I am not familiar with the topics. But some comments should be considered to improve the quality of the work. The comments are stated as follows:
- The introduction needs be enlarged, the background about the healthcare should be enhanced, the following papers should be cited: An Efficient Discrete Wavelet Transform Based Partial Hadamard Feature Extraction and Hybrid Neural Network Based Monarch Butterfly Optimization for Liver Tumor Classification, https://dx.doi.org/10.30919/es8d594; Acute-on-Chronic Liver Failure Mortality Prediction using an Artificial Neural Network, https://dx.doi.org/10.30919/es8d515; Ischemic Heart Disease Multiple Imputation Technique using Machine Learning Algorithm, https://dx.doi.org/10.30919/es8d681
- How can you link the topic to the membrane? What is the membrane used?
- What is the CPBs? Please add the full name.
Author Response
Dear Reviewer Thank you very much for your support and help, I expanded the background and explained what CPB is, specified the acronym CPBS, entered the type of oxygenator membrane used and entered the bibliographic references required in the manuscript (12-13- 14)
